# Signal Collapse in One-Shot Pruning: When Sparse Models Fail to Distinguish Neural Representations

## Abstract

The size of modern neural networks has made inference increasingly resource-intensive. Network pruning reduces model size by sparsifying parameters. One-shot pruning, which selects parameters via impact-based importance scores and applies second-order parameter updates, often incurs severe accuracy loss. We identify for the first time that this degradation occurs due to a phenomenon we refer to as *signal collapse*, which is a significant reduction in activation variance across layers, rather than the removal of 'important' parameters. To address this, we introduce **REFLOW**, which restores layer-wise activation variance without modifying any parameters. REFLOW uncovers high-quality sparse subnetworks within the original parameter space, enabling vanilla magnitude pruning to match or exceed complex baselines with minimal computational overhead. On ImageNet at 80% unstructured sparsity, REFLOW recovers ResNeXt-101 top-1 accuracy from below 0.41% to 78.9%, and at structured 2:4 N:M sparsity, it recovers ResNeXt-101 from 10.75% to 79.07%. By shifting the focus of the pruning paradigm from parameter selection to signal preservation, REFLOW delivers sparse models with state-of-the-art performance with minimal computational overhead.

## 1 Introduction

Modern neural networks comprise hundreds of millions to billions of parameters Young et al. (2017); Sung et al. (2024), making inference costly and often prohibitive in hardware-limited environments Rajbhandari et al. (2020). Pruning offers an efficient path by removing parameters while preserving accuracy Wang (2021); Jiang et al. (2022); Lee et al. (2019); Wang et al. (2020); Tanaka et al. (2020). In practice, many systems rely on *iterative pruning* by: (i) estimating *which weights to prune* at the current sparsity, (ii) removing them, (iii) fine-tuning the remaining weights to recover performance, and (iv) repeating until the target sparsity is reached. Each round requires retraining over large datasets; as sparsity grows, more rounds with careful pruning and learning rate schedules are needed. These prune–fine-tune–evaluate cycles scale poorly—for contemporary model sizes they often require days to weeks of compute Benbaki et al. (2023).

This motivates *one-shot* pruning: compress once, without retraining. One-shot methods fall into two categories: **magnitude pruning (MP)**, which removes small-magnitude weights Hanson & Pratt (1988); Mozer & Smolensky (1989); Han et al. (2015); Gordon et al. (2020), and **impact-based pruning (IP)**, which is loss-aware—estimating *weight importance* with *second-order (Hessian) information* to decide which parameters to prune, then applying a *single Hessian-based update on the surviving weights* to offset pruning-induced loss LeCun et al. (1989); Hassibi et al. (1993); Singh & Alistarh (2020); Benbaki et al. (2023). This one-shot Hessian-based update is *distinct from retraining*: there are no fine-tuning epochs; a single (approximate) second-order step adjusts the remaining parameters. While outperforming magnitude pruning, second-order information is costly because backward passes retain per-layer activations and intermediate tensors. Memory is dominated by activations (often exceeding the model weights) and well above that required by inference. Hessian-based estimates add further compute and memory overhead Singh & Alistarh (2020); Benbaki et al. (2023). In practice, for example, *Combinatorial Brain Surgeon, a one shot pruning method, takes hours to one-shot prune MobileNet* ($\approx$4.2M parameters) and does not scale well to larger architectures Yu et al. (2022).

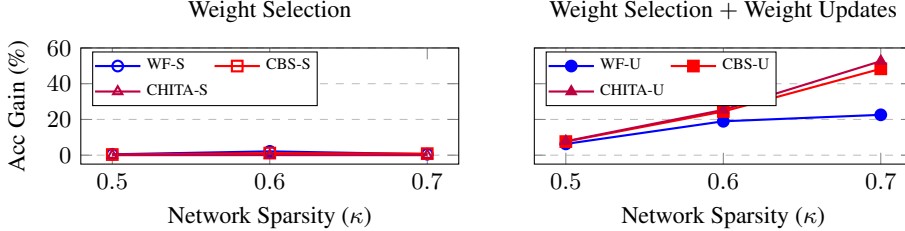

Figure 1: Comparison of test accuracy gain of impact-based pruning methods over magnitude pruning for a pre-trained MobileNet on ImageNet at different sparsity levels. **Left:** Selection-only pruning methods. **Right:** Pruning methods with weight updates achieve significant accuracy gains.

Empirically, *IP often outperforms MP in one shot* LeCun et al. (1989); Hassibi et al. (1993); Singh & Alistarh (2020); Benbaki et al. (2023). A straightforward hypothesis is that IP wins because it decides *which weights to prune* using loss-aware heuristic (gradients and Hessian information), whereas MP relies only on magnitude. We test this directly by decoupling IP into (i) *weight selection* and (ii) the *Hessian-based weight update*. When we keep only the selection step—WoodFisher-S, CBS-S, CHITA-S—performance matches MP (Figure 1, left), while the gains appear almost entirely *after* the Hessian-based update (Figure 1, right). Thus, differences in *which weights are pruned* are not the principal cause of the MP–IP performance gap.

**What, then, fails after pruning?** We identify a *new point of failure in one-shot pruning: signal collapse*. One-shot pruning *reduces activation variance at each layer*. This reduction activation variance cumulates across layers, resulting in nearly constant activations in the later layers (variance $\rightarrow 0$), so distinct inputs map to nearly identical representations, $\mathbf{f}(\theta, x_1) \approx \mathbf{f}(\theta, x_2)$ for $x_1 \neq x_2$. Crucially, IP's *single-shot Hessian-based weight update partially mitigates* this collapse by restoring activation variance—especially in early/mid layers—which explains much of IP's accuracy gain over MP despite similar weight selection.

Building on this observation, we introduce REFLOW, which *directly mitigates signal collapse by restoring activation variance*, enabling MP to outperform IP at high sparsity—*without* gradient/Hessian computations or weight updates. REFLOW runs end-to-end in a *few tens of seconds*, in sharp contrast to *hours* for second-order IP pipelines, and turns simple MP into a scalable baseline. On ImageNet at 80% sparsity, *ResNet-152* recovers from under 1% to **68.2%**, and *ResNeXt-101* from under 0.41% to **78.9%**, indicating that high-quality sparse models emerge by *restoring activation flow* rather than optimizing weight-selection heuristics, and that high-performing sparse sub-networks already exist *within* the original pre-trained weights.

**Contributions.** This work makes the following key observations and contributions:

1. For the first time in the context of pruning, we identify **signal collapse as the leading cause of accuracy loss in addition to the removal of critical weights**.

2. **Signal collapse can be mitigated without updating any trainable weights.** Our work REFLOW restores activation flow, enabling networks pruned by MP to outperform IP methods *without requiring gradient or Hessian computations*.

3. We demonstrate that **high-performing sparse sub-networks inherently exist in the original parameter space**. Unlike IP methods, which rely on updating unpruned weights to find a solution outside the original parameter space, our approach addresses signal collapse to uncover these sub-networks directly within the original weights.

## 2 BACKGROUND & RELATED WORK

This section provides the mathematical formulation of pruning and reviews existing work on pruning methods.

## 2.1 PROBLEM SETUP

Consider a pre-trained deep neural network (DNN) $f(\theta; x)$ parameterized by $\theta \in \mathbb{R}^d$ and input $x$. Pruning produces a sparse sub-network $f(\theta \odot m; x)$, where $m \in \{0,1\}^d$ is a binary mask, and $\odot$ denotes element-wise multiplication. Sparsity $\kappa \in [0,1]$ is the proportion of parameters set to zero. Pruning assigns scores $z \in \mathbb{R}^d$ to parameters importance, using methods ranging from simple weight magnitude to loss-aware based pruning scores.

## 2.2 RELATED WORK

**Magnitude-Based Pruning (MP)** is a simple and widely used pruning method Han et al. (2015); Frankle & Carbin (2019); Mozer & Smolensky (1989); Li et al. (2017); Tanaka et al. (2020); Renda et al. (2020); Gordon et al. (2020); Hanson & Pratt (1988); Liu et al. (2021); Eccles et al. (2024). MP ranks weights based on their absolute values:

$$z_i = |\overline{\theta}_i|. \tag{1}$$

It prunes parameters with the smallest magnitudes, which is computationally efficient. However, MP does not account for the impact of pruning on the loss function, which can result in suboptimal pruning decisions.

**Impact-Based Pruning (IP)** explicitly considers the loss function to guide pruning decisions LeCun et al. (1989); Hassibi & Stork (1992); Singh & Alistarh (2020). The impact of pruning is quantified as a second-order Taylor expansion of the loss function $\mathcal{L}$ centered at the pre-trained weights $\overline{\theta}$:

$$\mathcal{L}(\overline{\theta} + \delta\theta) - \mathcal{L}(\overline{\theta}) = \delta\theta^\top \nabla\mathcal{L}(\overline{\theta}) + \tfrac{1}{2}\delta\theta^\top H \delta\theta + O(\|\delta\theta\|^3), \tag{2}$$

where $H = \nabla^2 \mathcal{L}(\overline{\theta})$ is the Hessian.

Assuming $\overline{\theta}$ represents a local minimum of the loss (as is often the case for pre-trained networks), the gradient term $\nabla\mathcal{L}(\overline{\theta}) = 0$. For small perturbations $\delta\theta$, the higher-order terms become negligible, leading to the local quadratic approximation:

$$\mathcal{L}(\overline{\theta} + \delta\theta) - \mathcal{L}(\overline{\theta}) \approx \tfrac{1}{2}\delta\theta^\top H \delta\theta. \tag{3}$$

Below we review key IP methods that build on this quadratic approximation.

*Optimal Brain Damage (OBD)* improves on MP by estimating the increase in loss due to pruning LeCun et al. (1989). Assuming the Hessian $H$ is diagonal, the pruning score for a weight $\overline{\theta}_i$ is:

$$z_i = \frac{\overline{\theta}_i^2}{2H_{ii}}. \tag{4}$$

While OBD ranks weights based on their impact on loss using a diagonal Hessian approximation, it ignores parameter interactions.

*Optimal Brain Surgeon (OBS)* generalizes OBD by considering the full Hessian to capture cross-parameter interactions Hassibi et al. (1993):

$$z_i = \frac{\overline{\theta}_i^2}{2[H^{-1}]_{ii}}, \quad \delta\theta^* = \frac{-\overline{\theta}_i[H^{-1}]e_i}{[H^{-1}]_{ii}}. \tag{5}$$

Here, $z_i$ represents the pruning score, and $\delta\theta^*$ defines the Hessian-based weight updates applied to the unpruned weights. OBS is computationally expensive for modern networks due to the cost of inverting the Hessian $H$; nonetheless, it outperforms MP and OBD.

*Modern Hessian-Based Methods:* To reduce the computational cost of OBS, WoodFisher Singh & Alistarh (2020) introduces block-diagonal approximations of the Hessian via the empirical Fisher information matrix derived from a subset of training data:

$$H \approx \frac{1}{n} \sum_{i=1}^{n} \nabla\ell_i(\overline{\theta}) \nabla\ell_i(\overline{\theta})^\top, \tag{6}$$

where $\ell_i(\overline{\theta})$ is the loss for a single data point. This approximation reduces computational overhead but still focuses on pruning individual weights, without explicitly accounting for interactions between multiple weights.

*Pruning Multiple Weights:* Combinatorial Brain Surgeon (CBS) Yu et al. (2022) considers the joint effect of pruning multiple weights simultaneously, outperforming WoodFisher. However, its reliance on a dense Hessian $H \in \mathbb{R}^{p \times p}$ makes it computationally intensive, taking hours to prune MobileNet and is not scalable for large networks, such as ResNet-50. CHITA Benbaki et al. (2023) uses memory-efficient quadratic approximations for faster pruning than CBS but still relies on Hessian-based updates, modifying unpruned weights rather than identifying existing sparse sub-networks in the original parameter space.

## 3 REASSESSING IMPACT-BASED PRUNING

### 3.1 REVISITING WEIGHT SELECTION

As discussed above, MP selects weights based on their absolute magnitudes, while IP's weight selection leverages second-order approximations of the loss (see Equation 5), followed by Hessian-based weight updates. To isolate the effect of selection, we compare MP with 'selection-only' variants of IP (WF-S, CBS-S, CHITA-S), denoted as *IP-selection*, which prune without weight updates. We also include random pruning and vanilla MP as baselines.

Figure 2 (Left) shows that IP-selection (WF-S, CBS-S, CHITA-S) offers only marginal improvements (up to 2%) over MP, while random pruning severely reduces accuracy. This indicates that both MP and IP-selection identify meaningful parameters, unlike random pruning. However, the negligible difference between MP and IP-selection underscores the limited role of weight selection in pruning performance.

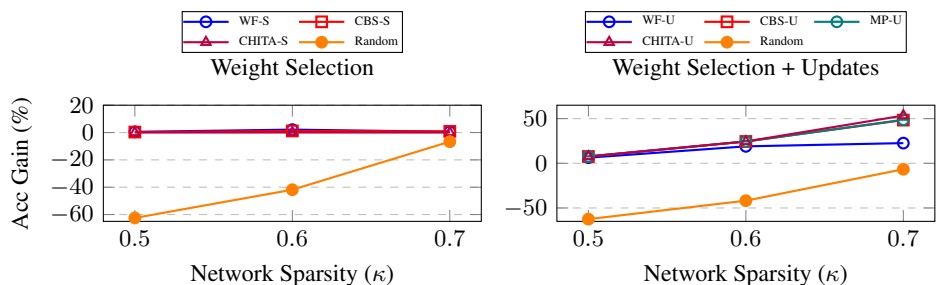

Figure 2: Comparison of test accuracy gain over magnitude pruning for a pre-trained MobileNet (trained on ImageNet) at different sparsity levels. **Left:** Selection-only pruning methods (IP-selection). **Right:** Methods with Hessian-based weight updates.

Further analysis of the similarity between pruning decisions made by MP and CHITA is provided in the Appendix E to demonstrate that both methods produce nearly identical masks, underscoring the limited role of weight selection.

### 3.2 ROLE OF HESSIAN-BASED WEIGHT UPDATES

While selection-only pruning methods have a limited impact on pruning performance, Hessian-based updates are critical for recovering accuracy after pruning. We therefore apply the same update step to MP (denoted MP-U), and compare it alongside the full IP methods with updates (WF-U, CBS-U, CHITA-U).

Figure 2 (Right) shows that MP-U achieves gains on par with WF-U, CBS-U, and CHITA-U—far outperforming selection-only and naive MP. This demonstrates that it is the Hessian-based update, not the choice of the pruning mask, that drives accuracy recovery. Combining MP's simple selection with Hessian updates matches the state of the art, making expensive IP-selection unnecessary.

**Insights**: Impact-based selection-only pruning offers limited gains over magnitude pruning, confirming the limited role of weight selection. In contrast, adding Hessian-based updates results in substantial accuracy recovery. These findings *shift the focus from weight selection to exploring other reasons*, beyond the pruning mask, that affect final pruning performance.

# 4 UNDERSTANDING SIGNAL COLLAPSE AND RESTORING PERFORMANCE LOSS WITH REFLOW

We examine why one-shot pruning resulting in severe performance loss by introducing *signal collapse* - a phenomenon in which activation variance vanishes in deep layers, rendering the network unable to distinguish inputs. We then present **REFLOW**, which restores variance without updating any trainable weights.

**Notation and Setup:** Consider a pretrained network $f(\theta)$ with parameters $\theta \in \mathbb{R}^d$. At layer $\ell$, let

$$\mathbf{X}_\ell \quad = \quad f(\mathbf{H}_{\ell-1}; \theta_\ell) \quad \text{and} \quad \mathbf{Z}_\ell(n) \quad = \quad \frac{\mathbf{X}_\ell(n) - \mu_\ell}{\sqrt{\mathrm{Var}_\ell^{(\mathrm{orig})}(\mathbf{X}_\ell) + \epsilon}} \gamma_\ell \quad + \quad \beta_\ell \quad (7)$$

denote the pre–BatchNorm activation and its BatchNorm output, respectively.

**Defining Signal Collapse:** Let $\mathrm{Var}_\ell^{(\mathrm{pruned})}$ and $\mathrm{Var}_\ell^{(\mathrm{orig})}$ be the post–BN variances at layer $\ell$ in the pruned and original networks. We say that *signal collapse* can be observed in a network if

$$\lim_{\ell \to L} \frac{\mathrm{Var}_\ell^{(\mathrm{pruned})}}{\mathrm{Var}_\ell^{(\mathrm{orig})}} \quad \to \quad 0, \tag{8}$$

where $L$ is the total number of layers. When the variance ratio approaches zero in deeper layers, the activations become nearly constant, producing uniform outputs, and the network thus loses its ability to distinguish between different inputs.

## 4.1 WHY PRUNING CAUSES SIGNAL COLLAPSE

Signal collapse originates from two complementary effects. First, pruning zeros out most weights and reduces the variance of the pruned pre–BN activation:

$$\mathrm{Var}_\ell^{(\mathrm{pruned})}(\mathbf{X}'_\ell) \quad \ll \quad \mathrm{Var}_\ell^{(\mathrm{orig})}(\mathbf{X}_\ell), \tag{9}$$

as shown in Appendix A.3.

Second, normalization operation in BatchNorm still divides by the original running variance, so the post–BN variance further reduces due to over-normalization:

$$\mathrm{Var}_\ell^{(\mathrm{pruned})}(\mathbf{Z}'_\ell) \quad \ll \quad \mathrm{Var}_\ell^{(\mathrm{orig})}(\mathbf{Z}_\ell). \tag{10}$$

See Appendix A.4 for further details.

## 4.2 CUMULATIVE REDUCTION IN ACTIVATION VARIANCE ACROSS LAYERS RESULTS IN SIGNAL COLLAPSE

We define the per–layer variance ratio as

$$\eta_\ell = \frac{\mathrm{Var}_\ell^{(\mathrm{pruned})}(\mathbf{Z}'_\ell)}{\mathrm{Var}_\ell^{(\mathrm{orig})}(\mathbf{Z}_\ell)} < 1.$$

Since each layer's input equals the previous layer's output ($\mathbf{H}_{\ell+1} = \mathbf{Z}_\ell$):

$$\mathrm{Var}_L^{(\mathrm{pruned})}(\mathbf{Z}'_L) = \left(\prod_{\ell=1}^{L} \eta_\ell\right) \mathrm{Var}_L^{(\mathrm{orig})}(\mathbf{Z}_L). \tag{11}$$

If $\eta_\ell \approx 0.9$ over $L = 25$ layers then

$$\prod_{\ell=1}^{25} 0.9 = 0.9^{25} \approx 0.072. \tag{12}$$

In the extreme, such that $\kappa \to 1$,

$$\lim_{\kappa \to 1} \mathrm{Var}_L^{(\mathrm{pruned})}(\mathbf{Z}'_L) = \left(\prod_{\ell=1}^{L} \eta_\ell\right) \mathrm{Var}_L^{(\mathrm{orig})}(\mathbf{Z}_L) \to 0. \tag{13}$$

**Insight:** Since $\mathrm{Var}(\mathbf{Z}'_L) \to 0$, the layer-$L$ outputs collapse to their mean,

$$\lim_{\mathrm{Var}(\mathbf{Z}'_L) \to 0} \mathbf{Z}'_L(n) = \mathrm{Mean}(\mathbf{Z}'_L), \tag{14}$$

so any two inputs map to nearly identical representations - $(\mathbf{Z}'_L(\mathbf{x}_1) \approx \mathbf{Z}'_L(\mathbf{x}_2))$.

### 4.3 EMPIRICAL VALIDATION

We empirically validate signal collapse via two **global scalar metrics** at each BN layer:

$$\mathrm{Mean}_\ell = \frac{1}{|\mathbf{Z}_\ell|} \sum_{x \in \mathbf{Z}_\ell} x, \quad \mathrm{Var}_\ell = \frac{1}{|\mathbf{Z}_\ell|} \sum_{x \in \mathbf{Z}_\ell} (x - \mathrm{Mean}_\ell)^2.$$

Figure 3 (Left) plots the ratio $\mathrm{Var}_\ell^{(\mathrm{pruned})}/\mathrm{Var}_\ell^{(\mathrm{orig})}$ at various sparsities $\kappa$, showing severe collapse for $\kappa = 0.9$. Figure 3 (Right) shows that 90%-sparse ResNet-20, which has undergone variance collapse, predicts almost all inputs to a single class. This behaviour aligns with our analysis of Equation 14, leading to nearly identical representations for different inputs.

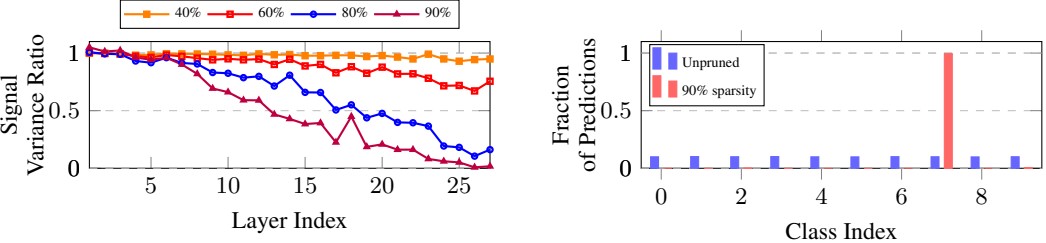

Figure 3: Signal collapse under high sparsity. **Left:** Layer-wise variance ratio $\mathrm{Var}_\ell^{(\mathrm{pruned})}/\mathrm{Var}_\ell^{(\mathrm{orig})}$ for MobileNet on ImageNet. Higher sparsity leads to signal collapse in deeper layers. **Right:** Class-prediction distribution of ResNet-20 on CIFAR-10, where the 90%-sparse model maps nearly all inputs to one class.

### 4.4 HESSIAN–BASED UPDATES MITIGATE SIGNAL COLLAPSE

Building on our earlier findings that Hessian-based weight updates are essential to recovering accuracy after pruning, we hypothesize that this is because they counteract signal collapse. In an 80%-sparse MobileNet on ImageNet, pruning with CHITA-S (without weight update) results in progressive variance collapse across layers, whereas CHITA-U (with weight update) mitigates complete signal collapse by recovering variance in the deeper layers. This confirms partial mitigation of variance collapse (see Figure 4).

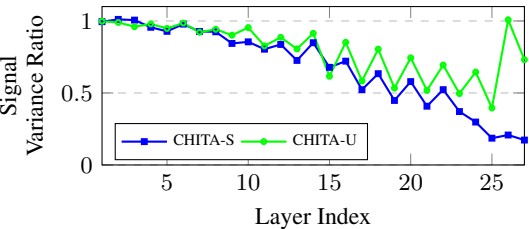

Figure 4: Layer-wise signal variance ratios $\frac{\mathrm{Var}^{(\mathrm{Pruned})}}{\mathrm{Var}^{(\mathrm{Baseline})}}$ in 80% sparse MobileNet on ImageNet.

### 4.5 RESTORING SIGNAL PROPAGATION TO MITIGATE COLLAPSE

To reverse collapse, we introduce **REFLOW** - a BN–recalibration that updates only each layer's running mean and variance (Appendix A.6). After pruning, we gather a small calibration set $\mathcal{B}$ and compute

$$\widehat{\mu}_\ell = \frac{1}{|\mathcal{B}|} \sum_{n \in \mathcal{B}} \mathbf{X}'_\ell(n), \qquad \widehat{\mathrm{Var}}_\ell = \frac{1}{|\mathcal{B}|} \sum_{n \in \mathcal{B}} \left(\mathbf{X}'_\ell(n) - \widehat{\mu}_\ell\right)^2. \tag{15}$$

Replacing the original BN statistics with the pruned-model statistics results in

$$\mathbf{Z}'^{(\mathrm{REFLOW})}_\ell(n) = \frac{\mathbf{X}'_\ell(n) - \widehat{\mu}_\ell}{\sqrt{\widehat{\mathrm{Var}}_\ell + \epsilon}} \, \gamma_\ell + \beta_\ell, \tag{16}$$

which fully restores the variance profiles to match the unpruned network (Figure 5) without updating any trainable weights.

## 5 EXPERIMENTAL RESULTS

We apply REFLOW to magnitude pruning (MP) and evaluate it across small, medium, and large architectures. The results highlight that REFLOW consistently recovers performance in pruned networks, achieving state-of-the-art accuracy without requiring computationally expensive Hessian-based updates. By mitigating signal collapse, REFLOW discovers high-quality sparse subnetworks within the original parameter space.

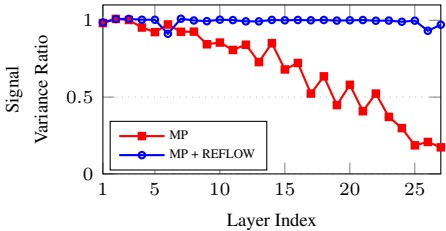

Figure 5: Variance ratios in pruned MobileNet (ImageNet) at 80% sparsity.

### 5.1 PERFORMANCE ON SMALL ARCHITECTURES

We begin by evaluating REFLOW on small architectures, namely ResNet-20 He et al. (2015) pre-trained on CIFAR-10 Krizhevsky (2009) and MobileNet Howard et al. (2017) pre-trained on ImageNet Deng et al. (2009), with less than 5 million parameters and comparing them to state-of-the-art one-shot pruning methods, namely WF Singh & Alistarh (2020), CBS Yu et al. (2022), and CHITA Benbaki et al. (2023).

Table 1 highlights REFLOW's accuracy improvements across all sparsity levels. For ResNet-20, REFLOW restores accuracy to 49.16% at 0.9 sparsity, outperforming CHITA (15.60%) and MP (11.79%). On MobileNet, REFLOW achieves 43.37% accuracy at 0.8 sparsity, surpassing CHITA (29.78%) and MP (0.11%).

Table 1: Performance of pruning methods on small architectures (ResNet-20 on CIFAR-10; MobileNet on ImageNet) at various sparsities. Unpruned accuracies are 91.57% and 71.96%. Best results in **bold**.

| Dataset | Network | Sparsity | MP | WF | CBS | CHITA | REFLOW |
|---|---|---|---|---|---|---|---|
| CIFAR-10 | ResNet-20 | 0.4 | 89.98 | 91.15 | 91.21 | 91.19 | **91.25** |
| | | 0.5 | 88.44 | 90.23 | 90.58 | 90.60 | **90.66** |
| | | 0.6 | 85.24 | 87.96 | 88.88 | 89.22 | **89.49** |
| | | 0.7 | 78.79 | 81.05 | 81.84 | 84.12 | **86.65** |
| | | 0.8 | 54.01 | 62.63 | 51.28 | 57.90 | **78.50** |
| | | 0.9 | 11.79 | 11.49 | 13.68 | 15.60 | **49.16** |
| ImageNet | MobileNet | 0.4 | 69.16 | 71.15 | 71.45 | 71.50 | **71.59** |
| | | 0.5 | 62.61 | 68.91 | 70.21 | 70.42 | **70.48** |
| | | 0.6 | 41.94 | 60.90 | 66.37 | 67.30 | **67.83** |
| | | 0.7 | 6.78 | 29.36 | 55.11 | 59.40 | **61.54** |
| | | 0.8 | 0.11 | 0.24 | 16.38 | 29.78 | **43.37** |
| Weight Update | - | - | ✗ | ✓ | ✓ | ✓ | ✗ |

## 5.2 SCALING REFLOW TO MEDIUM-SIZED ARCHITECTURES

We evaluate REFLOW on medium-sized architectures, namely ResNet-50 pre-trained on ImageNet (25 million parameters). For this size, we compare REFLOW to CHITA and M-FAC Frantar et al. (2021), as WF and CBS are computationally prohibitive. Figure 6 shows that REFLOW outperforms CHITA and M-FAC across all sparsity levels. At high sparsities, REFLOW offers superior accuracy without the overhead of Hessian computation.

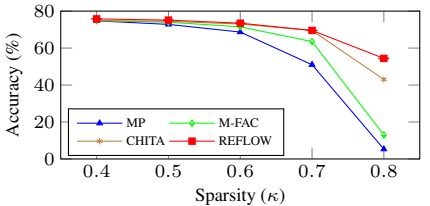

Figure 6: ResNet-50 test accuracy vs. network sparsity on ImageNet.

## 5.3 SCALING REFLOW TO LARGE ARCHITECTURES

To test REFLOW's scalability, we prune four large ImageNet models ($\geq$100M parameters) - ResNet-101, ResNet-152, RegNetX-32GF, and ResNeXt-101 at 80% sparsity. Impact-based methods cannot cope at this scale: CBS relies on computing a dense Hessian, and CHITA requires multiple gradient passes per layer, making them impractical for large networks. Vanilla magnitude pruning collapses below 5% accuracy on all models, whereas REFLOW recovers Top-1 accuracies of 64.1%, 68.2%, 73.0%, and 78.9%, respectively. In particular, on ResNeXt-101, REFLOW restores accuracy from 0.4% to 78.9%, just 4.0% below the dense 82.9% baseline despite removing 80% of weights. These results further demonstrate that signal collapse is a fundamental bottleneck in one-shot pruned networks.

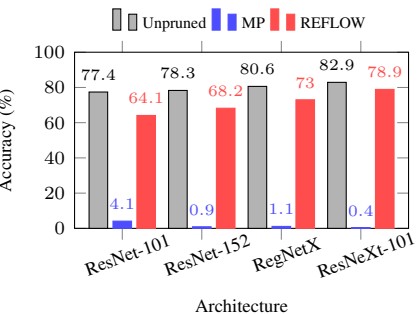

Figure 7: Unpruned, MP, and REFLOW ImageNet accuracy at 80% sparsity on various architectures.

## 5.4 EXTENSION TO STRUCTURED SPARSITY

We extended our evaluation to structured pruning patterns, specifically 2:4 (N:M) sparsity, which offers practical hardware speedups. Note that existing one-shot Hessian-based methods (WoodFisher, CBS, CHITA) do not natively support structured sparsity. Table 2 reports ImageNet top-1 accuracies for ResNet-50 and ResNeXt-101 under 2:4 structured sparsity, comparing magnitude pruning (MP) with and without REFLOW.

Table 2: Structured Sparsity (2:4) on ImageNet (inference speed-up relative to dense model).

| Model | Baseline | MP | REFLOW (speedup) |
|---|---|---|---|
| ResNet-50 | 76.13% | 4.28% | 64.03% (1.3×) |
| ResNeXt-101 | 82.99% | 10.75% | 79.07% (1.2×) |

## 5.5 COMPARISONS IN THE CONTEXT OF GRADUAL PRUNING

We compare REFLOW extensively against prior *gradual* (prune–retrain) methods on **ResNet-50/ImageNet**. For a fair setup, we use the **STR** pre-trained checkpoint and the **Incremental** polynomial sparsity schedule. At each pruning step we apply magnitude pruning followed by REFLOW as a fast, forward-only calibration (no gradients/Hessians). Table 3 reports pruned top-1 accuracy at **80%** and **90%** sparsity, showing that REFLOW (gradual) is competitive with state-of-the-art gradual pruning baselines.

Table 3: Pruned top-1 accuracy on ImageNet (**ResNet-50**) under gradual pruning with light retraining baselines. REFLOW (gradual) applies REFLOW after each pruning step (forward-only).

| Sparsity | GMP+LS | VD | RIGL+ERK | SNFS+LS | STR | DNW | REFLOW |
|---|---|---|---|---|---|---|---|
| 0.80 | 75.58 | 75.28 | 75.10 | 74.90 | 76.19 | 76.20 | 76.60 |
| 0.90 | 73.91 | 73.84 | 73.00 | 72.90 | 74.31 | 74.00 | 75.09 |

## 5.6 Convergence with REFLOW

Building on the results in Table 1, we evaluate the impact of REFLOW across pruning methods with varying complexities: MP, CHITA-S (selection-only), and CHITA (selection with Hessian-based updates). CHITA updates the unpruned weights using second-order information, while CHITA-S applies the same selection criteria without weight updates. This distinction isolates the role of weight updates and quantifies whether REFLOW can compensate for their absence.

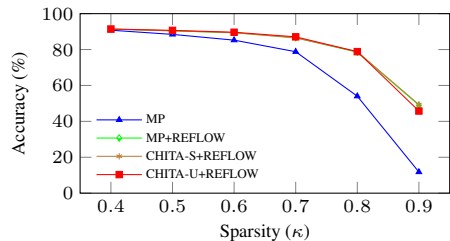

Figure 8: ResNet-20 test accuracy vs. Network sparsity on CIFAR-10.

Figure 8 shows that REFLOW bridges the performance gap between MP, CHITA-S, and CHITA-U. REFLOW enables simpler selection based approaches, such as MP and CHITA-S, to achieve comparable accuracy as CHITA-U (Hessian-based weight updates), although the latter is computationally intensive. This highlights that mitigating signal collapse, rather than employing complex pruning selection heuristics, is the key to recovering performance in one-shot pruned networks.

## 5.7 Extension to Transformer Architectures

We also observe *signal collapse* in Transformers. In CNNs, BatchNorm exposes running means/variances that we can recalibrate after pruning, whereas LayerNorm computes statistics *per sample* and exposes only the affine parameters $(\gamma_\ell, \beta_\ell)$. We therefore briefly recalibrate these LN parameters on a small calibration set while freezing all other weights, which restores activation variance and recovers large accuracy drops (Table 4). Extending this analysis to LLMs is future work; Appendix A.7 gives the method and derivations.

Table 4: ImageNet pruning of ViT variants: accuracy with magnitude pruning (MP) vs. MP+LN update. Baselines: ViT-B/16 81.07%, ViT-L/32 76.96%.

| Model | ViT-B/16 | | | | ViT-L/32 | | | |
|---|---|---|---|---|---|---|---|---|
| Sparsity | 0.4 | 0.5 | 0.6 | 0.7 | 0.4 | 0.5 | 0.6 | 0.7 |
| MP | 54.87 | 26.50 | 6.74 | 0.47 | 57.85 | 34.49 | 8.95 | 0.94 |
| MP+LN | 77.65 | 75.51 | 71.33 | 62.62 | 72.57 | 69.86 | 65.53 | 58.64 |

## 6 Conclusion

This work identifies signal collapse as a critical bottleneck in one-shot neural network pruning. Performance loss in pruned networks is due to **signal collapse** in addition to the removal of critical parameters. We propose **REFLOW** (**Re**storing **F**low of **Low**-variance signals), a simple yet effective method that mitigates signal collapse without computationally expensive weight updates. REFLOW highlights the importance of mitigating signal collapse in sparse networks and enables magnitude pruning to match or surpass state-of-the-art one-shot pruning methods such as CHITA, CBS, and WF.

REFLOW consistently achieves state-of-the-art accuracy across diverse architectures, restoring ResNeXt-101 from under 0.41% to 78.9% top-1 accuracy at 80% sparsity on ImageNet. Its lightweight design makes it a practical solution for delivering high-quality sparse models without the overhead of traditional approaches. These findings challenge the traditional emphasis on weight selection strategies and underscore the critical role of maintaining signal propagation for achieving high-quality sparse networks in the context of one-shot pruning.

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

## A FULL DERIVATIONS FOR SIGNAL COLLAPSE AND REFLOW

All notation used in Section 4 is highlighted and complete, self-contained proofs of the key inequalities and equations (Equation (9) – Equation (16)) are presented.

### A.1 DEFINITIONS AND ASSUMPTIONS

At layer $\ell$ of the original network (cf. Equation (7)):

$$\mathbf{X}_\ell = f(\mathbf{H}_{\ell-1}; \theta_\ell), \quad \mathbf{Z}_\ell(n) = \frac{\mathbf{X}_\ell(n) - \mu_\ell}{\sqrt{\mathrm{Var}_\ell^{(\mathrm{orig})}(\mathbf{X}_\ell) + \epsilon}} \gamma_\ell + \beta_\ell. \tag{17}$$

Here $\mathbf{H}_{\ell-1} = \mathbf{Z}_{\ell-1}$ is the post–BN output of layer $\ell - 1$, and $(\mu_\ell, \mathrm{Var}_\ell^{(\mathrm{orig})}(\mathbf{X}_\ell))$ are BN's stored running mean and variance.

**Probabilistic assumptions.** For each fixed batch index $n$, we assume the components of $\mathbf{H}_{\ell-1}(n) = (H_{\ell-1,1}(n), \ldots, H_{\ell-1,d}(n))$ satisfy

$$\mathbb{E}[H_{\ell-1,i}(n)] = 0, \quad \mathrm{Cov}(H_{\ell-1,i}(n), H_{\ell-1,j}(n)) = 0 \quad (i \neq j). \tag{18}$$

These zero-mean and uncorrelated assumptions are standard in pruning and BatchNorm analyses.

### A.2 WEIGHT MASKING NOTATION

After one-shot pruning at sparsity $\kappa$, we zero most weights. We define

$$\mathcal{S} = \{ i : W'_{\ell,i} \neq 0 \}, \quad W'_{\ell,i} = \begin{cases} W_{\ell,i}, & i \in \mathcal{S}, \\ 0, & \text{otherwise.} \end{cases} \tag{19}$$

Hence the pruned pre–BN activation is

$$\mathbf{X}'_\ell(n) = \sum_{i \in \mathcal{S}} W'_{\ell,i} H_{\ell-1,i}(n). \tag{20}$$

### A.3 PRUNED PRE–BN VARIANCE (PROOF OF EQUATION (9))

We now prove that

$$\mathrm{Var}_\ell^{(\mathrm{pruned})}(\mathbf{X}'_\ell) \ll \mathrm{Var}_\ell^{(\mathrm{orig})}(\mathbf{X}_\ell), \tag{21}$$

i.e. main Equation (9).

**Proof.** From Equation (20) we have

$$\mathbf{X}'_\ell(n) = \sum_{i \in \mathcal{S}} W'_{\ell,i} H_{\ell-1,i}(n).$$

By the zero-mean assumption in Equation (18),

$$\mathbb{E}[\mathbf{X}'_\ell(n)] = \sum_{i \in \mathcal{S}} W'_{\ell,i} \mathbb{E}[H_{\ell-1,i}(n)] = 0.$$

Hence, by definition of variance,

$$\mathrm{Var}[\mathbf{X}'_\ell(n)] = \mathbb{E}[(\mathbf{X}'_\ell(n) - 0)^2] = \mathbb{E}\Big[\Big(\sum_{i \in \mathcal{S}} W'_{\ell,i} H_{\ell-1,i}(n)\Big)^2\Big].$$

Expanding the square inside the expectation gives two terms:

$$\Big(\sum_{i \in \mathcal{S}} W'_{\ell,i} H_{\ell-1,i}\Big)^2 = \sum_{i \in \mathcal{S}} (W'_{\ell,i})^2 H_{\ell-1,i}^2 + \sum_{\substack{i,j \in \mathcal{S} \\ i \neq j}} W'_{\ell,i} W'_{\ell,j} H_{\ell-1,i} H_{\ell-1,j}.$$

Taking expectations term by term and using $\mathbb{E}[H_{\ell-1,i}] = 0$ and $\mathrm{Cov}(H_{\ell-1,i}, H_{\ell-1,j}) = 0$ for $i \neq j$,

$$\mathbb{E}[H_{\ell-1,i}^2] = \mathrm{Var}[H_{\ell-1,i}], \qquad \mathbb{E}[H_{\ell-1,i}H_{\ell-1,j}] = 0 \quad (i \neq j).$$

Thus

$$\mathrm{Var}\big[\mathbf{X}'_\ell(n)\big] = \sum_{i \in \mathcal{S}} (W'_{\ell,i})^2 \, \mathbb{E}[H_{\ell-1,i}^2] \; + \; \sum_{i \neq j} W'_{\ell,i} W'_{\ell,j} \, \underbrace{\mathbb{E}[H_{\ell-1,i}H_{\ell-1,j}]}_{0}$$

$$= \sum_{i \in \mathcal{S}} (W'_{\ell,i})^2 \, \mathrm{Var}[H_{\ell-1,i}].$$

On the other hand, the unpruned activation $\mathbf{X}_\ell(n) = \sum_{i=1}^{d} W_{\ell,i} H_{\ell-1,i}(n)$ has variance

$$\mathrm{Var}_\ell^{(\mathrm{orig})}(\mathbf{X}_\ell) = \sum_{i=1}^{d} W_{\ell,i}^2 \, \mathrm{Var}[H_{\ell-1,i}].$$

Since $\mathcal{S} \subset \{1, \dots, d\}$ and $|\mathcal{S}| \ll d$, dropping most nonnegative summands gives

$$\sum_{i \in \mathcal{S}} W_{\ell,i}^2 \, \mathrm{Var}[H_{\ell-1,i}] \; \ll \; \sum_{i=1}^{d} W_{\ell,i}^2 \, \mathrm{Var}[H_{\ell-1,i}],$$

which completes the proof of Equation (21).

### A.4 Over–Normalization by BatchNorm (Proof of Equation (10))

We next prove

$$\mathrm{Var}_\ell^{(\mathrm{pruned})}(\mathbf{Z}'_\ell) \; \ll \; \mathrm{Var}_\ell^{(\mathrm{orig})}(\mathbf{Z}_\ell), \tag{22}$$

i.e. main Equation (10).

**Proof.** Even after pruning, BN still uses its stored $\mu_\ell$ and $\sigma_\ell^2 = \mathrm{Var}_\ell^{(\mathrm{orig})}(\mathbf{X}_\ell)$:

$$\mathbf{Z}'_\ell(n) = \frac{\mathbf{X}'_\ell(n) - \mu_\ell}{\sqrt{\sigma_\ell^2 + \epsilon}} \, \gamma_\ell + \beta_\ell.$$

Adding $\beta_\ell$ is shift–invariant, so $\mathrm{Var}(\mathbf{Z}'_\ell) = \mathrm{Var}\big((\mathbf{X}'_\ell - \mu_\ell)\gamma_\ell / \sqrt{\sigma_\ell^2 + \epsilon}\big)$. By $\mathrm{Var}(aX + b) = a^2 \mathrm{Var}(X)$:

$$\mathrm{Var}_\ell^{(\mathrm{pruned})}(\mathbf{Z}'_\ell) = \left(\frac{\gamma_\ell}{\sqrt{\sigma_\ell^2 + \epsilon}}\right)^2 \mathrm{Var}_\ell^{(\mathrm{pruned})}(\mathbf{X}'_\ell), \tag{23a}$$

$$\mathrm{Var}_\ell^{(\mathrm{orig})}(\mathbf{Z}_\ell) = \left(\frac{\gamma_\ell}{\sqrt{\sigma_\ell^2 + \epsilon}}\right)^2 \mathrm{Var}_\ell^{(\mathrm{orig})}(\mathbf{X}_\ell). \tag{23b}$$

Dividing Equation (23a) by Equation (23b) and using Equation (21) gives Equation (22).

### A.5 Cumulative Collapse Across Layers (Proof of Equation (11) – Equation (13))

Define the per-layer factor

$$\eta_\ell = \frac{\mathrm{Var}_\ell^{(\mathrm{pruned})}(\mathbf{Z}'_\ell)}{\mathrm{Var}_\ell^{(\mathrm{orig})}(\mathbf{Z}_\ell)}, \quad 0 < \eta_\ell < 1. \tag{24}$$

Since $\mathbf{H}_{\ell+1} = \mathbf{Z}_\ell$, one shows by induction:

$$\mathrm{Var}_{\ell+1}^{(\mathrm{pruned})}(\mathbf{X}'_{\ell+1}) = \eta_\ell \, \mathrm{Var}_{\ell+1}^{(\mathrm{orig})}(\mathbf{X}_{\ell+1}) \tag{25}$$

and therefore

$$\mathrm{Var}_L^{(\mathrm{pruned})}(\mathbf{Z}'_L) = \left(\prod_{\ell=1}^{L} \eta_\ell\right) \mathrm{Var}_L^{(\mathrm{orig})}(\mathbf{Z}_L), \tag{26}$$

with $\prod_{\ell=1}^{L} \eta_\ell \to 0$ as $\kappa \to 1$, yielding

$$\lim_{\kappa \to 1} \mathrm{Var}_L^{(\mathrm{pruned})}(\mathbf{Z}'_L) = 0. \tag{27}$$

### A.6 REFLOW CALIBRATION (PROOF OF EQUATION (16))

Collect a small calibration set $\mathcal{B}$ of size $B$ and compute

$$\widehat{\mu}_\ell = \frac{1}{B} \sum_{n \in \mathcal{B}} \mathbf{X}'_\ell(n), \tag{28a}$$

$$\widehat{\mathrm{Var}}_\ell = \frac{1}{B} \sum_{n \in \mathcal{B}} \left(\mathbf{X}'_\ell(n) - \widehat{\mu}_\ell\right)^2. \tag{28b}$$

Replace each BN layer's stored $(\mu_\ell, \sigma_\ell^2)$ by $(\widehat{\mu}_\ell, \widehat{\mathrm{Var}}_\ell)$. Then

$$\mathbf{Z}'^{(\mathrm{REFLOW})}_\ell(n) = \frac{\mathbf{X}'_\ell(n) - \widehat{\mu}_\ell}{\sqrt{\widehat{\mathrm{Var}}_\ell + \epsilon}}\, \gamma_\ell + \beta_\ell, \tag{29}$$

exactly matching main Equation (16). By construction, $\mathrm{Var}[\mathbf{Z}'^{(\mathrm{REFLOW})}_\ell] = \mathrm{Var}^{(\mathrm{pruned})}_\ell(\mathbf{X}'_\ell)/\widehat{\mathrm{Var}}_\ell = 1$ (up to $\epsilon$), fully restoring the variance profile.

**Summary of Assumptions.** All proofs rely on (i) zero-mean, uncorrelated pre–BN activations Equation (18), (ii) $\mathbb{E}[\mathbf{X}'_\ell] = 0$ after masking, and (iii) fixed BN running statistics until recalibration—standard in second-order pruning analyses and sufficient to explain—and correct—signal collapse via REFLOW.

### A.7 EXTENSION TO TRANSFORMER ARCHITECTURES

**Motivation.** One-shot pruning in Transformers also results in signal collapse: layer-wise activation variance contracts with depth, leading to *signal collapse* and severe accuracy loss. Unlike BatchNorm (BN), which exposes running $(\mu_\ell, \sigma_\ell^2)$ for post-pruning recalibration, LayerNorm (LN) computes statistics *per sample* and exposes only affine parameters $(\gamma_\ell, \beta_\ell)$—so variance restoration must act through these parameters rather than recomputing global moments.

**Calibration budget and sample efficiency.** For ViTs, we recalibrate only LN affine parameters using a small labeled calibration set of $\approx 500$ mini-batches, which in our setup takes $\leq 5$ minutes wall clock. With batch size 128, that corresponds to $500 \times 128 = 64{,}000$ images—about **5%** of ImageNet's 1.28M training images. Consistent with our CNN results, we also observed that accuracy saturates quickly with far fewer batches (e.g., tens of batches suffice in the BN-recalibration setting), underscoring that the calibration acts as variance restoration rather than full fine-tuning.

**Notation and LN–affine calibration.** Let $X'_\ell(n) \in \mathbb{R}^{d_\ell}$ be the post-pruning pre-LN activation at layer $\ell$ for example $n$. LN produces

$$Z'_\ell(n; \gamma, \beta) = \frac{X'_\ell(n) - \mu_n(X'_\ell)}{\sqrt{\mathrm{Var}_n(X'_\ell) + \epsilon}}\, \gamma + \beta, \tag{30}$$

where $\gamma, \beta \in \mathbb{R}^{d_\ell}$ are elementwise affine parameters. We recalibrate *only* $(\gamma_\ell, \beta_\ell)$ by minimizing

$$(\gamma_\ell^*, \beta_\ell^*) = \arg\min_{\gamma, \beta} \frac{1}{|\mathcal{B}|} \sum_{n \in \mathcal{B}} \mathcal{L}\big(f(Z'_\ell(n; \gamma, \beta)), y_n\big), \tag{31}$$

freezing all other weights (attention, MLP, embeddings). A few hundred forward-backwards passes over $\mathcal{B}$ with a first-order optimizer (e.g., Adam) is sufficient because the number of calibrated parameters is tiny relative to the full model.

**Why it mitigates collapse.** Pruning shrinks pre-LN variance. Although LN enforces unit variance *pre-affine*, the downstream effective scales and centers are governed by $(\gamma_\ell, \beta_\ell)$. Calibrating these parameters restores appropriate activation magnitudes and recenters features across depth, halting the compounding attenuation that yields near-constant late-layer representations.

**Empirical outcomes (ImageNet).** As reported in Table 4 (main body), LN–affine calibration converts large MP collapses into strong accuracies for both ViT-B/16 and ViT-L/32; e.g., at 60% sparsity MP yields 6.74%/8.95% vs. 71.33%/65.53% after calibration, with similarly large gains from 40–70% sparsity. These trends parallel our BN recalibration results for CNNs, indicating that activation-variance preservation is the key driver of post-pruning recovery regardless of architecture.

## B  DETAILED COMPARISON WITH REPAIR

REPAIR (Jordan et al., 2023) addresses a variance collapse that arises when two pre-trained networks are linearly interpolated. Denote their aligned layer-$\ell$ pre-BatchNorm activations on input $n$ by $\mathbf{X}_\ell^{(1)}(n)$ and $\mathbf{X}_\ell^{(2)}(n)$. Form the convex interpolation

$$\mathbf{X}_{\ell,\alpha}(n) = (1-\alpha)\,\mathbf{X}_\ell^{(1)}(n) + \alpha\,\mathbf{X}_\ell^{(2)}(n), \quad \alpha \in [0,1]. \tag{32}$$

By bilinearity of variance, one obtains

$$\mathrm{Var}\big[\mathbf{X}_{\ell,\alpha}\big] = (1-\alpha)^2\,\mathrm{Var}\big[\mathbf{X}_\ell^{(1)}\big] + \alpha^2\,\mathrm{Var}\big[\mathbf{X}_\ell^{(2)}\big] + 2\,\alpha(1-\alpha)\,\mathrm{Cov}\big[\mathbf{X}_\ell^{(1)}, \mathbf{X}_\ell^{(2)}\big]. \tag{33}$$

Let $\sigma_i = \sqrt{\mathrm{Var}[\mathbf{X}_\ell^{(i)}]}$. Since $\mathrm{Cov}[\mathbf{X}_\ell^{(1)}, \mathbf{X}_\ell^{(2)}] \leq \sigma_1\sigma_2$, the interpolated variance is strictly less than the squared convex combination,

$$\mathrm{Var}[\mathbf{X}_{\ell,\alpha}] < \big((1-\alpha)\sigma_1 + \alpha\sigma_2\big)^2.$$

REPAIR restores the intended standard deviation $(1-\alpha)\sigma_1 + \alpha\sigma_2$ by inserting a temporary BatchNorm layer with scale $\beta$ satisfying

$$\beta\,\sqrt{\mathrm{Var}[\mathbf{X}_{\ell,\alpha}]} = (1-\alpha)\,\sigma_1 + \alpha\,\sigma_2, \tag{34}$$

which yields the closed-form

$$\beta = \frac{(1-\alpha)\,\sigma_1 + \alpha\,\sigma_2}{\sqrt{(1-\alpha)^2\,\sigma_1^2 + \alpha^2\,\sigma_2^2 + 2\,\alpha(1-\alpha)\,\mathrm{Cov}[\mathbf{X}_\ell^{(1)}, \mathbf{X}_\ell^{(2)}]}}. \tag{35}$$

In contrast, REFLOW traces collapse to one-shot pruning in a single network. A pruning mask $m_{\ell,i}$ reduces the pre-BatchNorm variance $\mathrm{Var}[\mathbf{X}_\ell]$ by dropping weight contributions, yielding $\mathrm{Var}[\mathbf{X}_\ell'] \ll \mathrm{Var}[\mathbf{X}_\ell]$ and hence a post-BN ratio $\eta_\ell < 1$ that compounds across layers (Equation (11)–Equation (13)). REFLOW then gathers a small calibration set $\mathcal{B}$ and recomputes each layer's running moments $(\mu_\ell, \sigma_\ell^2)$ via empirical estimates $\widehat{\mu}_\ell, \widehat{\mathrm{Var}}_\ell$ (Equation D.6 – Equation D.7), producing the corrected activation

$$\mathbf{Z}_\ell'^{(\mathrm{REFLOW})}(n) = \frac{\mathbf{X}_\ell'(n) - \widehat{\mu}_\ell}{\sqrt{\widehat{\mathrm{Var}}_\ell + \epsilon}}\,\gamma_\ell + \beta_\ell,$$

which by construction restores $\mathrm{Var}[\mathbf{Z}_\ell]$ exactly (Equation (16)).

Although both methods employ an affine variance-restoration, REPAIR's $\beta$ depends on two-network variances and their covariance (Equation 35), whereas REFLOW's recalibration relies solely on the pruned model's own statistics and a brief calibration. These differences in context, dependencies, and derivation underscore that REFLOW is the first weight-update-free, alignment-free solution for activation variance collapse in one-shot pruning.

## C  EXPERIMENTAL SETUP

This section provides a detailed overview of the experimental setup used in our study, including the pruning techniques, datasets, sparsity ranges, and computational environment.

We employed a range of established one-shot pruning techniques, which perform pruning in a single step, followed by Hessian-based updates of the remaining weights and reduce the impact on loss after pruning. Specifically, we considered WoodFisher Singh & Alistarh (2020), CBS Yu et al. (2022),

CHITA Benbaki et al. (2023), and Matrix-Free Approximate Curvature (M-FAC) Frantar et al. (2021). Performance metrics for these methods were sourced from existing literature Yu et al. (2022); Benbaki et al. (2023), with results averaged over three independent runs.

**Application of REFLOW:** In this work, REFLOW is applied to networks pruned using *magnitude pruning*. After pruning, Batch Normalization (BN) running statistics are recalibrated using a forward pass over a limited number of training samples.

**Hyperparameters:** For REFLOW, we used 50 training batches to recalibrate the running BN statistics, with a batch size of 128 across all experiments.

**Pre-Trained Networks and Datasets:** To ensure comparability with prior studies Yu et al. (2022); Benbaki et al. (2023), we adopted datasets and model architectures from the same studies. The analysis included three pre-trained networks: ResNet-20 He et al. (2015) trained on the CIFAR-10 dataset Krizhevsky (2009), and MobileNet Howard et al. (2017) and ResNet-50 He et al. (2015) trained on the ImageNet dataset Deng et al. (2009).

We extended the analysis to include larger architectures that prior leading one-shot pruning methods Singh & Alistarh (2020); Yu et al. (2022) did not explore and are unable to scale to efficiently. Specifically, we evaluated REFLOW on ResNet-101 He et al. (2015), ResNet-152 He et al. (2015), RegNetX Radosavovic et al. (2020), and ResNeXt-101 Xie et al. (2017), all trained on the ImageNet dataset.

**Sparsity Range:** We evaluated REFLOW across the following sparsity ranges, consistent with prior works Yu et al. (2022); Benbaki et al. (2023):

- **ResNet-20 on CIFAR-10:** Sparsity range of 0.4 to 0.9.
- **MobileNet on ImageNet:** Sparsity range of 0.4 to 0.8.
- **ResNet-50 on ImageNet:** Sparsity range of 0.4 to 0.9.

**Hardware:** All experiments were conducted on a computational setup comprising an NVIDIA RTX A6000 GPU with 48GB memory, 10,752 CUDA cores, and 336 Tensor cores capable of 309 TFLOPS peak performance, coupled with an AMD EPYC 7713P 64-Core CPU.

**Software:** The computational environment operated on Ubuntu 20.04.6 LTS (Focal Fossa), utilizing Python version 3.8.10 and PyTorch version 2.1.0.

## D ABLATION STUDIES

In this section, we evaluate the performance of REFLOW through ablation studies. We analyze the impact of the number of training batches ($N$), layer-wise BN recalibration, and batch size on accuracy recovery in pruned networks.

### D.1 EFFECT OF THE NUMBER OF TRAINING BATCHES ON PERFORMANCE

We analyze the impact of varying the number of training batches ($N$) on the performance of RE-FLOW, focusing on test accuracy. REFLOW is applied to sparse networks after magnitude pruning, recalibrating Batch Normalization (BN) statistics through a forward pass over $N$ training batches.

Figure 9 shows the relationship between $N$ and test accuracy for MobileNet at 80% sparsity. Accuracy improves significantly for small values of $N$, saturating around $N = 50$. Using $N = 50$ training batches with a batch size of 128 corresponds to only 6,400 images—less than 0.5% of the 1.28 million training samples in ImageNet.

In contrast, leading impact-based pruning methods such as WoodFisher Singh & Alistarh (2020) and CBS Yu et al. (2022) require 960,000 training samples for gradient computation, while CHITA Benbaki et al. (2023) requires 16,000 samples. REFLOW achieves comparable performance using just 6,400 samples without any gradient computation, relying solely on forward passes to update BN statistics. This minimal data requirement enables REFLOW to operate in scenarios where access to the full training dataset is limited, such as privacy-preserving applications or resource-constrained environments, where re-training is infeasible.

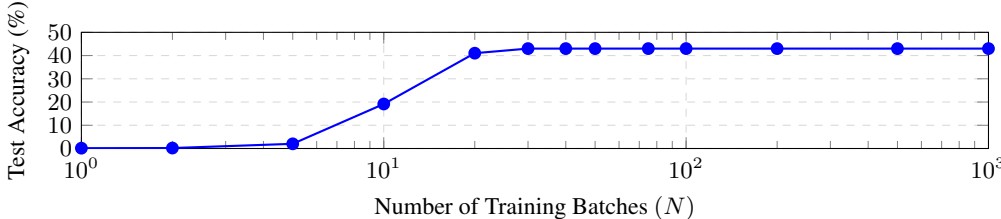

Figure 9: Test accuracy of MobileNet at 80% sparsity using REFLOW for different numbers of training batches ($N$). Accuracy improves significantly for $N \leq 20$, saturates around $N = 50$, and stabilizes for larger $N$.

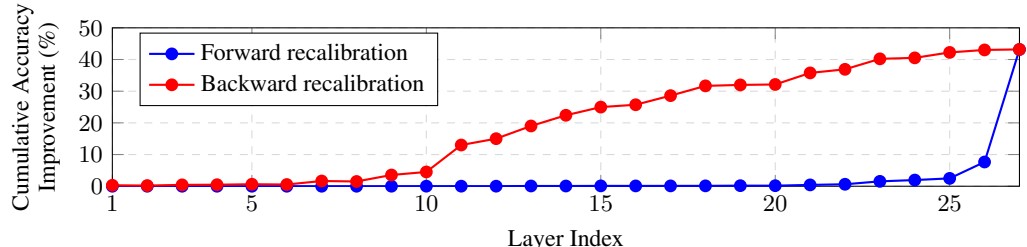

Figure 10: Cumulative accuracy improvement (%) for MobileNet at 80% sparsity after one-shot magnitude pruning. Forward recalibration progresses from the first BN layer to the last, while backward recalibration starts from the last BN layer. Backward recalibration achieves significant improvements earlier than forward recalibration, reflecting the higher sensitivity of deeper layers to pruning-induced changes.

### D.2  IMPACT OF LAYER-WISE RECOVERY ON PERFORMANCE

To gain deeper insights into the recovery of test accuracy in sparse networks, we analyzed the contribution of individual Batch Normalization (BN) layers by recalibrating them sequentially. Specifically, the recalibration was performed one layer at a time, measuring the cumulative improvement in test accuracy after recalibrating each BN layer. This process was conducted in two directions: from the first BN layer to the last (forward direction) and from the last BN layer to the first (backward direction).

Figure 10 presents the cumulative effect of BN recalibration on test accuracy for MobileNet at 80% sparsity after one-shot pruning. In the forward direction, recalibrating early BN layers contributes minimally to accuracy recovery, with notable improvements only emerging as deeper layers are recalibrated. This pattern suggests that the shallower layers are less sensitive to changes in their BN statistics, whereas deeper layers play a more critical role in preserving network performance. Conversely, in the backward direction, recalibrating late BN layers produces substantial accuracy gains early on, with diminishing returns as earlier layers are recalibrated. These observations indicate that later layers are disproportionately impacted by pruning-induced changes, reflecting their higher sensitivity.

This behavior aligns with the phenomenon of *signal collapse*, where the variance of activations diminishes significantly in deeper layers of the pruned network. The variance ratio between pruned and original activations approaches zero in the final layers, leading to near-constant activations. This results in indistinguishable representations, which propagate to the output, causing uniform or incorrect predictions. The pronounced recovery observed when recalibrating the last layers supports this theoretical insight: correcting the BN statistics in these layers mitigates signal collapse, restoring the discriminative power of the network's activations.

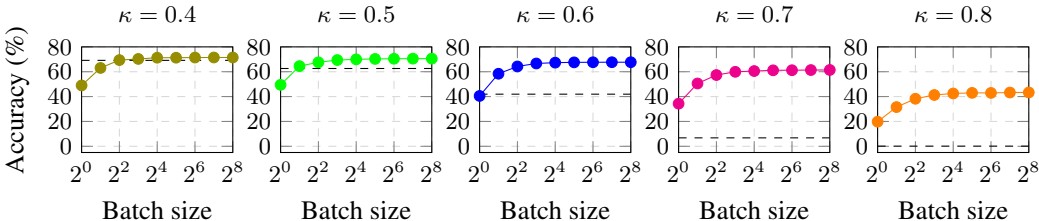

Figure 11: Test accuracy of MobileNet at different sparsity levels ($\kappa$) and varying batch sizes on ImageNet using REFLOW. Dashed lines represent the baseline accuracy for Magnitude Pruning (MP) without REFLOW.

### D.3    EFFECT OF BATCH SIZE ON PERFORMANCE

Here, we investigate the influence of varying batch sizes on the test accuracy of REFLOW for different target sparsity levels ($\kappa$) as shown in Figure 11.

At lower sparsity levels ($\kappa = 0.4$ and $\kappa = 0.5$), using smaller batch sizes for REFLOW results in a drop in accuracy below the baseline performance of Magnitude Pruning (MP). This indicates that insufficient recalibration data can negatively impact performance in less sparse networks. However, increasing the batch size leads to a noticeable improvement in accuracy, with REFLOW surpassing MP at moderate and large batch sizes. These results demonstrate that networks with lower sparsity still benefit from recalibration when sufficient batch statistics are available.

For intermediate sparsity ($\kappa = 0.6$), the impact of batch size is more pronounced. Accuracy improves consistently with larger batch sizes, significantly outperforming MP even at smaller batch sizes. Saturation occurs at moderate batch sizes, highlighting the increased dependency on recalibration as network sparsity increases.

At higher sparsity levels ($\kappa = 0.7$ and $\kappa = 0.8$), larger batch sizes are critical for achieving substantial gains over MP. Accuracy improves steadily with batch size, with saturation occurring at higher batch sizes compared to lower sparsity levels. These results highlight the importance of recalibration in mitigating the performance degradation caused by high sparsity. The dashed lines in Figure 11 provide a reference to the baseline MP performance, underscoring the effectiveness of REFLOW in recovering accuracy, particularly for highly sparse networks.

## E    ANALYZING PRUNING SIMILARITY USING HAMMING DISTANCE

To further understand the limited role of weight selection, we analyze the *Normalized Hamming Distance* between pruning masks produced by MP, CHITA, and random pruning. CHITA is used as the representative state-of-the-art (SOTA) IP method.

The *Hamming Distance* between two masks $m^{(A)}$ and $m^{(B)}$ is defined as:

$$H(m^{(A)}, m^{(B)}) = \sum_{i=1}^{d} \mathbb{I}\left(m_i^{(A)} \neq m_i^{(B)}\right),$$

where $\mathbb{I}(\cdot)$ is the indicator function, $d$ is the total number of parameters, and $m_i = 1$ indicates that parameter $i$ is retained. The *Normalized Hamming Distance*, which measures the fraction of differing pruning decisions between two masks, is defined as:

$$H_{\text{norm}}(m^{(A)}, m^{(B)}) = \frac{H(m^{(A)}, m^{(B)})}{d}.$$

where $H(m^{(A)}, m^{(B)})$ is the Hamming Distance, and $d$ is the total number of parameters.

Figure 12 shows that the Normalized Hamming Distance between MP and CHITA is negligible, indicating close similarity in their pruning decisions compared to the significant variation with random pruning. For ResNet-20 on CIFAR-10, it is $0.0018\%$. For MobileNet on ImageNet, it is $0.0095\%$. These results show that magnitude-based and IP-selection methods make nearly identical pruning

decisions, supporting the conclusion that the choice of weight selection (MP or IP-selection) has minimal influence on pruning performance.

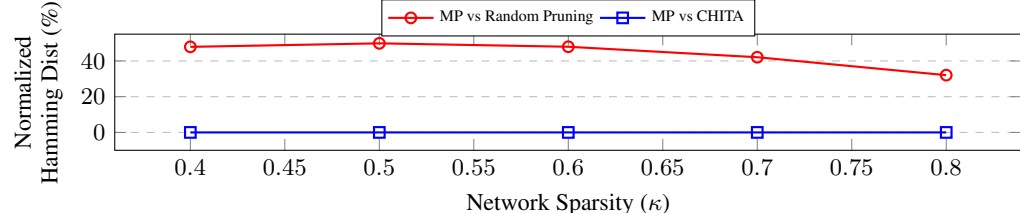

Figure 12: Normalized Hamming Distance (%) between pruning masks for Magnitude Pruning (MP) vs Random pruning and MP vs CHITA across sparsity levels. MP and CHITA have negligible variation, while MP and Random pruning show significant differences.

