# OpenReview forum: "Signal Collapse in One-Shot Pruning: When Sparse Models Fail to Distinguish Neural Representations"
_ICLR.cc/2026/Conference — ICLR 2026 Conference Withdrawn Submission_

### Official Review · Reviewer_Lwa5 · 2025-10-24

**Soundness:** 3
**Presentation:** 3
**Contribution:** 2
**Rating:** 6
**Confidence:** 3

**Summary:**

The authors start by showing empirically that pruning methods are generally worse than pruning followed by compensation, for recovering generalization performance of sparse networks. They highlight that one of the potential explanations for this behaviour is the variance collapse for neurons after sparsification. This collapse seems to be partially mitigated by compensation after pruning. The authors present Reflow to recalibrate this variance by adjusting the running mean and variance of normalization layers after pruning leading to significant gains in performance.

**Strengths:**

Strengths

1. The authors highlight a problem that exists in pruning and training, i.e., variance collapse.
2. They further show that this can be mitigated via calibration of normalization params.
3. Experiments on ViTs and CNNs highlight the effectiveness of the method.

**Weaknesses:**

Weaknesses

1. The authors show that variance collapse can explain the poor performance after pruning. However, I believe this paper would benefit from delving into the details of why this is the case. In ViTs, this phenomenon does not directly seem to translate and instead the learnable LN params are recalibrated. This suggests that the problem might also be in the learnable BN or LN params, but the analysis for this is missing. ([1] might be relevant for this).
2. Instead of recalibrating the running means, have the authors considered rescaling it wrt the sparsity of the neuron, like [2], this might be faster.
3. Distentangling the effects of the running stats from the learnable params of normalization might also allow extending the method to LLMs where this problem also exists if i understand correctly.

Overall, this method shows significant gain in performance by mitigating variance collapse, but it would make this paper much stronger if the variance collapse was studied in more detail as outlined above.


[1] Oh, Junghun, et al. "Batch normalization tells you which filter is important." Proceedings of the IEEE/CVF Winter Conference on Applications of Computer Vision. 2022.

[2] Lasby, Mike, et al. "Dynamic sparse training with structured sparsity." arXiv preprint arXiv:2305.02299 (2023).

**Questions:**

In addition, the comparison of REFLOW is with impact based pruning methods, but the comparison with the compensation based methods is missing? Does REFLOW outperform these methods too?

The running stats of BN are only used for inference, how does this affect the training dynamics?

---

### Official Review · Reviewer_vFHD · 2025-10-25

**Soundness:** 4
**Presentation:** 2
**Contribution:** 2
**Rating:** 4
**Confidence:** 4

**Summary:**

The paper try to address the accuracy loss observed during one shot pruning, identifying the core issue as signal collapse which is a significant reduction in activation variance across model layers which makes sparse models fail to distinguish neural representations. The authors demonstrate this phenomenon by showing theoretically and empirically that layer-wise activation ratios diminish progressively in deeper layers.  The authors propose the solution, REFLOW, a batchnorm recalibration method that mitigates signal collapse by restoring layer-wise activation variance.

**Strengths:**

The authors are proposing an insight that overturns the conventional result. It was taken for granted that impact based pruning is better than magnitude pruning. Conventionally, metrics used in impact based pruning are considered better than simple weight magnitude in magnitude pruning in existing works. However, this paper refutes the existing results with the experiments. Below are the elements how this paper tries this.
- Mathematically proving the diminishing variance and its effect over the layers: The authors proposed and proved the problem that pruning leads to diminishing variance over the layers and it results identical representations on different inputs.
- Displaying the diminishing variance problem and its recovery with experiments: The authors support their claim with experimental results utilizing signal variance ratio. Diminishing variance problem is verified with diminishing signal variance ratio. The effect of hessian-based update of existing works and REFLOW are shown with recovered signal variance ratio.

While proposing the problem and supporting with mathematical proof and small experiment are clear, I still have large concerns on proposed results.

**Weaknesses:**

I integrated the weaknesses points into Section:Qeustions.

**Questions:**

1. The axis of Figure 2 is hard to identify the exact gain of IP-selections and MP. Since the existing works are insisting their effect with 1~2% accuracy gain, this figure is not enough to propose certain insight that IP selection offers limited gains over MP selection.
2. How’s the result when REFLOW is applied on IP selection without weight update? Is there any reason that the base selection method should be MP?
3. Proposed accuracy results are not enough to prove the effect of proposed result. The models tested are quite a bit old model (Mobilenet v1) and comparison with other methods is limited in Imagenet scale of Mobilenet v1 and ResNet50 (this is more limited than Mobilenet).

---

### Official Review · Reviewer_TPGG · 2025-10-25

**Soundness:** 2
**Presentation:** 3
**Contribution:** 1
**Rating:** 0
**Confidence:** 4

**Summary:**

The paper analyses reduction in activation variance across layers after pruning and develops REFLOW a simple method, which just runs batchnorm recalibration after pruning.

**Strengths:**

The paper is easy to read, and the method works.
Activation variance reduction analysis is sound.

**Weaknesses:**

The REFLOW method (aka Batchnorm tuning) is too simplistic and was proposed before, e.g., in the Optimal brain compression (https://arxiv.org/pdf/2208.11580) paper, which mentions Batchnorm tuning on pages 8/9.

Also, batchnorm is not used in Transformer models, which are the most common ones these days, so the method is not applicable here.

**Questions:**

See weaknesses.

---

### Official Review · Reviewer_4i49 · 2025-10-31

**Soundness:** 1
**Presentation:** 2
**Contribution:** 1
**Rating:** 2
**Confidence:** 4

**Summary:**

Post-training pruning degrades models performance, and authors claim that the main cause of the failure is signal collapse–a phenomena where variance of activations collapse at deeper layers with high sparsity. To address this, authors suggest REFLOW, a lightweight post-training pruning procedure which re-calibrates BN statistics to mitigate signal collapse with a small set of calibration data to improve performance.

**Strengths:**

- clear diagnosis attempt on post-training pruning failure: identifies and studies signal collapse as a post-training pruning failure mode (variance reduction in deeper layers w/ higher sparsity)
- suggests REFLOW: simple, cost-efficient yet performant technique to improve post-training pruning

**Weaknesses:**

- Lack of novelty: BN tuning –which tunes batch-norm statistics after pruning/quantization– is a known, widely used post-hoc correction [1,2,3]. REFLOW seems to be identical to BNT.
- Restricted application: REFLOW is only applicable to BN-based architectures. Although authors demonstrate that REFLOW is effective in LayerNorm-based architecture (Table 4), it is unclear whether ‘signal collapse’ actually occurs there and how the analysis extends.
- Missing baselines: although impact-based pruning methods (WF,CHITA,CBS) are included, connectivity(sensitivity)-based approaches, which has been studied extensively in pruning at initialization to preserve signal propagation [4,5,6], are missing.
- Over-strong assumptions: In Equation 18 of Appendix A.1, The “post-BN zero-mean and uncorrelated” assumption seems to be over-strong for pretrained, pruned networks. For instance, the BN operator itself does not whiten(decorrelates) activations [7], and pruned weights $(w’)$ may shift the activation distribution, possibly violating the zero-mean assumption for the next layer.

### References
[1] Hubara et al., Accelerated sparse neural training: A provable and efficient method to find n: m transposable masks. NeurIPS, 2021.
[2] Frantar and Alistarh, Optimal brain compression: A framework for accurate post-training quantization and
pruning. NeurIPS, 2022.
[3] Lee et al., SAFE: Finding Sparse and Flat Minima to Improve Pruning. ICML, 2025.
[4] Lee et al., SNIP: Single-shot Network Pruning based on Connection Sensitivity. ICLR, 2019
[5] Tanaka et al., Pruning neural networks without any data by iteratively conserving synaptic flow. NeurIPS, 2020
[6] Wang et al., Picking winning tickets before training by preserving gradient flow. ICLR, 2020.
[7] Huang et al., Decorrelated Batch Normalization. CVPR, 2018

**Questions:**

- REFLOW vs. BNT: What is the difference between REFLOW and standard BN tuning used after pruning/quantization?
- Justification of zero-mean/uncorrelated assumptions: Please either provide references that this assumption is standard, or demonstrate that pre-trained, pruned BN networks do have zero-mean and uncorrelated pre-activations.
- Signal collapse in different architectures: Does signal collapse occur in layer-norm based architectures? would the signal-collapse analysis naturally extend to these architectures?
- Signal-preserving baselines: How does REFLOW compare to SNIP/GraSP/SynFlow? Does REFLOW still help when applied on top of these criteria?
- add proper references for section 5.5-iterative pruning methods (GMP, RIGL, SNFS, STR, DNW)
- add compute cost of other methods in section 5.3 (e.g., reference to Appendix D)

---

### Note · Authors · 2025-11-17

I have read and agree with the venue's withdrawal policy on behalf of myself and my co-authors.